# Sources of evidence, pathways, and processes to support disability-inclusive decision-making in low- and middle-income countries: A scoping review

Ebruphiyo Ruth Useh [1]*, Zara Trafford[1], Prince Changole[1], Xanthe Hunt[1,2,3]

1 Institute for Life Course Health Research, Department of Global Health, Stellenbosch University, Cape Town, Western Cape, South Africa, 2 Africa Health Research Institute (AHRI), Somkhele, KwaZulu-Natal, South Africa, 3 Mental health, Alcohol, Substance Use, and Tobacco Research Unit (MAST-RU), South African Medical Research Council (SAMRC), Cape Town, South Africa

* Useh@sun.ac.za

## Abstract

Widespread failure to sufficiently account for disability in policies and programming across sectors limits the inclusion of people with disabilities in global health and development strategies. These oversights are especially marked in low- and middle-income countries (LMICs), where 80% of the world's 1.3 billion people with disabilities reside. To bridge disability policy and programming gaps, it is important to first understand how key categories of decision-makers think about disability in their work, including which sources of evidence, processes, and pathways they rely on to make policy and programming decisions about disability, and what influences (aside from evidence) shape their decision-making. To address these issues, we conducted a scoping review of the literature concerning the use of evidence in governmental and non-governmental decision-making around disability-inclusion in LMICs. We systematically searched databases of peer-reviewed literature and used thorough hand searches to gather grey literature. Documents were eligible if they focused on key governmental and non-governmental stakeholders; disability-inclusive or disability-related decision-making in mainstream or targeted planning, policy-making, programming, or evaluations; and were based on data from LMICs. All literature was double screened and extracted according to a standardised extraction sheet by four reviewers, working in pairs. We included 16 papers, with sources of evidence cited being highly variable and encompassing both empirical and experiential evidence, while barriers and facilitators to using evidence varied by evidence source. Outside of evidence, notable influences on decision-making included government legislation, power dynamics and involvement, stakeholder relationships, the local landscape, funding, and attitudes towards disability. This work highlights the barriers to, and enablers of, evidence utilisation in relation to disability, which can be targeted with intervention

**Data availability statement:** The data underlying this research is available in our data repository on the Open Science Framework (OSF) at https://doi.org/10.17605/OSF.IO/2Y4ST).

**Funding:** This scoping review was funded in full by Wellspring Philanthropies (grant ID: 17737; PI: XH). The funders had no role in study design, data collection and analysis, the decision to publish, nor the preparation of this manuscript. Funder URL: https://wpfund.org/

**Competing interests:** The authors have declared that no competing interests exist.

and advocacy. Moreover, this review suggests that supporting evidence-based decision-making in relation to disability in LMICs necessitates engagement with varied framings of evidence and influences on decision-making outside of evidence.

## Introduction

Although about 1.3 billion people worldwide have a disability, individuals with disabilities consistently experience poorer access to, and worse outcomes in, education, employment, community participation and health [1,2]. At least in part, this results from the continued neglect of disability in policies, legislation, services and interventions in planning and programming globally [3–5].

As a field, global health is centrally concerned with a multisectoral understanding of the forces shaping health and wellbeing. The field's focus on the social determinants of health necessitates this expansive gaze, and engagement with issues [6] beyond healthcare systems [6–8]. Global health moves beyond public health and medicine to attend to the role of policy, regulation, and structural inequalities in all sectors, and their impact on health and wellbeing.

Similarly, while disability has historically been situated within healthcare research, contemporary perspectives demonstrate the importance of a multisectoral lens, examining intervention, policy and service delivery for people with disabilities across health, education, and social development [7]. Both global public health scholarship, and disability research, then, highlight the need for public health to engage with the broader determinants of health, such as education, geographical location, and socioeconomic status, and the multiple sectors concerned with them [9–12]. This review showcases how disability—as a policy and programming topic which sits across health and other sectors—can be the site of multisectoral understanding and analysis, and shows how topics can be recognised as both key concerns for scholars of global health, as well as a cross-sectoral issue requiring a comprehensive and inclusive approach.

Over the last two decades, attention to 'evidence-based decision-making' has grown in the fields concerned with international development [13–15]. Goldman and Pabari [16] describe the development of policy and programming as taking place in stages, namely: agenda-setting, diagnosis of the issue at hand, selection of the intervention that can be used to address the issue, design or planning of the policy or programme, implementation, assessment and continued learning based on the outcome and impact of the policy or programme. At various points in the policy cycle or the development of a programme, evidence can be used to strengthen decision-making.

Despite broad endorsement of data-driven decision-making, policymakers may still rely on intuition, crises, and political pressures to make decisions [17]. And while governmental information systems are increasingly designed to increase evidence utilisation in decision-making, their effective implementation and uptake remains limited in many under-resourced settings [18]. Yet, in low- and middle-income countries

(LMICs), evidence-based decision-making is a vital component of optimised resource use, improved population health and wellbeing, and achievement of milestones in relation to international consensus frameworks like the Sustainable Development Goals (SDGs) [19].

Evidence can also be used to construct a theory of change, identify relevant outcomes, and define appropriate indicators of scale or quality in the design of an intervention, whether it is a policy or programme [15]. Additionally, evidence can be utilised as part of monitoring during implementation to evaluate progress, and essentially, for evaluating impact [15].

The intersection of disability-inclusion and evidence-based decision-making is rarely explored outside clinical contexts, creating a significant gap in the literature. Firstly, the notion of 'evidence' in diverse decision-making scenarios remains unclear, and advocacy for evidence-based approaches often privileges Western, empirical data. Non-empirical forms of evidence, such as those produced by actively involving people with disabilities (as recommended by the United Nations Convention on the Rights of Persons with Disabilities (UNCRPD)) [20]. Yet, whether such consultations are considered part of evidence-based decision-making, remains uncertain [21].

Secondly, existing literature often associates the inattention to disability in policies and programmes across all sectors [22] in LMICs to the lack of local evidence to inform such decision-making [23,24]. Regardless of what is considered evidence, there is usually a shortage of it that is specific to people with disabilities and their needs and experiences in LMICs, partly because studies on the impact of interventions for people with disabilities are most commonly conducted in high-income countries (HICs) [3]. Demographic data such as types of disability, longer term health outcomes, and disaggregation by disability status is used to inform even the most fundamental planning by LMICs. However, this kind of data often does not provide the level of information necessary to address a matter, or provides information that is not uniformly recognised across definitions or contexts of disability [25]. More specifically, disaggregation by disability status in larger datasets is understood to present as a challenge [26,27].

Thirdly, there is a general absence of formalised processes or documented best practices for drawing on evidence—even different types of evidence—to inform decision-making. This, coupled with the wide range of forces influencing the work of policy-makers and programme developers and managers, complicates efforts to strengthen the use of evidence in decision-making processes. Importantly, in LMICs, structural factors such as resource constraints, social norms, and post-colonial global power relations influence policy-making and programming landscapes in ways that may [28,29] not fit as neatly with the evidence-based decision-making models developed in HIC contexts. Qualitative studies by researchers working in policy and implementation studies in LMICs [30–33] suggest that factors like trust, relationships, funding agendas and specific constellations of power, among other things, are prominent in shaping the development and implementation of policies and programmes. Furthermore, even when key stakeholders advocate for the use of evidence in the design and implementation of LMIC policies and strategies [5], there is a lack of guidance on how best to do this in relation to disability. Current approaches used by researchers, such as policy briefs, oral presentations, and reports, may be of limited impact, or their impact may be more significantly shaped by factors outside of the specific dissemination strategy, such as relationships, networks, and decision-maker interests. So, there is also a need for innovative approaches for communicating evidence to those who have the power to make decisions.

Each of the issues detailed above requires clarification. Some, such as what constitutes reliable evidence, may not be readily amenable to resolution because they relate to epistemology. Others though, such as how to improve *utilisation* of different types of evidence by key stakeholders, may be answerable through engagement with existing literature.

This review begins with an exploration of the extent to which the existing literature documents how disability-related decisions in policy and programming are made in LMICs. We also wanted to understand what information policy-makers and programme managers use to make decisions, what other influences there might be on their decision-making, and what processes are used to make decisions. This scoping review aims to fill this knowledge gap by mapping and consolidating any literature which documents the sources of evidence, pathways, and processes among key governmental and non-governmental stakeholders (i.e., policy-makers and programme managers) involved in disability-inclusive decision-making in LMICs.

## Materials and methods

### Research questions

**Primary research question(s).** The primary research question for this review was: "What is the nature of the literature concerning the use of evidence in governmental and non-governmental decision-making around disability-inclusion (i.e., what types of literature exist, where is this work published, what are its key concerns, and what kinds of key stakeholders are included)?"

**Secondary research question(s).** This scoping review focused on examining existing evidence in the public domain with the intention of answering the following specific sub-questions:

1) What sources of information/evidence do key stakeholders draw on to inform their disability-relevant decisions in policy and programming?

2) Aside from information/evidence, what kinds of factors drive decision-making in disability-relevant policy and programming?

3) What are the barriers to, and facilitators of, evidence-based decision-making for LMIC governmental and non-governmental stakeholders in policy and programming in sectors including health, education, livelihoods, empowerment, and social inclusion?

### Data sources and search strategy

An extensive review was undertaken to describe and analyse findings from a wide range of published literature, including both peer-reviewed and grey sources. This review was conducted according to Mak and Thomas' scoping review methodology derived from Arksey and O'Malley's scoping review seminal paper, which involves five key steps [34]. In the first step, the research question is identified. In the second step, the relevant studies are identified. In the third step, the studies are selected for review. In the fourth step, the data is recorded. In the fifth step, the results are assembled, summarised and reported. A sixth additional step is the option of stakeholder consultations [34].

For this, our inclusion criteria were

- *Sample*: Key stakeholders, particularly policy-makers and programme managers working in various sectors such as health, education, social development and so forth, who make high-level decisions in all types of disability policy or programming

- *Phenomenon of interest*: Papers that focus on how key stakeholders, who make high-level decisions, went about making a decision or a set of decisions about disability policy or programming.

In this paper, we draw on the definition of disability as outlined in the UNCRPD which defines disability as a "long-term physical, mental, intellectual or sensory impairment, which, in interaction with various barriers, may hinder [a person's] full and effective participation in society on an equal basis with others" [20, p. 4]. Impairments include those which are sensory, physical, intellectual, related to self-care or communication, or are psychosocial. However, when we undertook the formative work for this scoping review, we recognised that psychosocial impairments, which are often attended to under the umbrella of mental health rather than disability, have seen investment in, consideration of, and publishing about, policy-making which has not been mirrored in relation to the concerns of people with sensory, physical, intellectual, self-care, or communication impairments (for instance, in respect of rehabilitation, assistive technology, or early identification). As such, in this analysis, we limit our inquiry to literature concerning disability but excluding psychosocial impairments/mental health.

This is not to imply that the authors do not consider psychosocial disability to be a form of disability, but rather because it is not possible to meaningfully distinguish between literature focused on mental health *but not psychosocial disability*. Therefore, all mental health literature was considered together. We observed the literature on mental health to be far

larger than the literature concerning people with sensory, physical, intellectual, self-care, or communication impairments. The concerns of people with sensory, physical, intellectual, self-care, or communication impairments may differ, in some systematic ways, from those documented in the global mental health literature, and so lessons learned from the latter literature may have limited applicability to the concerns of people with disabilities, generally.

- *Design*: All types of research methodologies used to collect primary research evidence exploring disability and decision-making in LMICs

- *Evaluation*: Paper's that focused on decision-making on disability-related matters—sources of information policy-makers and programme managers use to make decisions, other influences on their decision-making, and what processes are used to make decisions

- *Research type*: grey literature reports; peer-reviewed studies; qualitative, quantitative, or mixed-method studies exploring disability and decision-making that reported primary research evidence exploring disability and decision-making in LMICs

  Our exclusion criteria were as follows

- Any papers that focused solely on HICs, were unrelated to disability, or did not concern high-level decision-making (e.g., respondents with disabilities describing access issues, limitations on their participation, difficulties at home, income problems, etc.) were excluded.

- Literature published before the year 1990 was not included in this scoping review. This is because we understand evidence-based decision-making to have gained recognition among medical professionals around this time [1].

According to the above methodology, we firstly conducted thorough searches of various databases of peer-reviewed material. The selection of databases were based on the senior author's expert insight and extensive experience in disability and in review methods. These databases included Cumulative Index to Nursing and Allied Health Literature (CINAHL), Education Resources Information Center (ERIC), Scopus, Web of Science Social Sciences Citation Index, Medical Literature Analysis and Retrieval System Online (MEDLINE(R)), Excerpta Medica Database (Embase) Classic+Embase, PsycINFO, Cochrane, and Commonwealth Agricultural Bureaux (CAB) Global Health. Next, we conducted thorough online searches to gather relevant grey literature using the websites of the following large agencies and organisations: United Nations Educational, Scientific and Cultural Organization (UNESCO), World Bank, International Labour Organisation (ILO), World Health Organization (WHO), United Nations Children's Fund (UNICEF), SightSavers, Christian Blind Mission (CBM), International, Disability Alliance, United Nations High Commissioner for Refugees (UNHCR), Humanity and Inclusion, Inclusion International, Global Policy Forum, Save the Children, World Vision International, International Rescue Committee (IRC) and Catholic Relief Services (CRS), as well as supplementary Google Scholar searches. To improve the reporting and methodological quality of this review, we used the Preferred Reporting Items for Systematic Reviews and Meta-Analyses Extension for Scoping Reviews (PRISMA-ScR) checklist (please see S1 File) and employed a comprehensive search strategy to mitigate for bias (please see S2 File).

In the hand search of the grey literature, we selected relevant articles based on our inclusion and exclusion criteria. For our grey literature and Google Scholar hand search, the first 3 tabs/pages were searched using search strings (please see S3 File). This is because search results yielded past the first page 3 became less specific and did not satisfy our inclusion criteria. All searches were conducted between the 27th of August 2023 and the 14th of December 2023. The registered protocol for this scoping review is available at https://doi.org/10.17605/OSF.IO/D78HY.

## Study selection process

During study selection, the third step of the scoping review process, the authors used Rayyan.ai, a software platform specifically developed to facilitate collaborative evidence reviews in teams. Working in two pairs, we carefully assessed

and selected literature based on our predetermined inclusion criteria, double screening all titles and abstracts. At the end of the full text article-screening phase, the reviewers reported a disagreement rate of 33%. These disagreements were resolved by the senior author.

### Data extraction

For step four of this scoping review, all relevant information from each record was compiled in a spreadsheet. Information was extracted based on our research questions and included various details such as the authors' names and publication year, publication title, study design, types of literature, category of participants, level or ambit of decision-making, sectors and topics covered, impairment or health condition of focus, setting, data collection methods, sample size, aims, and a summarised study findings section. In addition, we prepared a more specific study findings section(s) that considered sources of evidence used, barriers and facilitators to using these sources of evidence, and other influences on disability-inclusive decision-making. Three reviewers assembled the data into a spreadsheet which was used to double-extract all articles, where the third reviewer (first author) extracted 100% of the articles, which were then refined (by the second author). Any disagreements or uncertainties were resolved during team meetings, also involving the fourth (senior) author.

### Data analysis

In the fifth and final step of this scoping review, we conducted data analysis that included numeric and qualitative content analyses. Qualitative content analysis was used to create codes and synthesise non-numerical data. Codes were created from inductively developed, condensed units of meaning using steps as described by Erlingsson and Brysiewicz [35]. Numeric analysis focused on quantitatively summarising five sections: the aims of papers, sources of evidence used in decision-making, influences on decision-making aside from evidence, as well as the barriers to, and facilitators of, using the aforementioned sources of evidence in decision-making. To ensure methodological rigour, analysis was completed by three of the authors, who developed analytical codes for the five sections mentioned above and documented emerging patterns. The final codes were validated by a fourth member of the research team, with additional discussions on codes held until consensus was reached.

We draw from Davies' [36] classification of evidence in the results to categorise the sources of evidence consulted in disability-related decision-making identified within the literature. Cairney [37] defines evidence as a claim or argument supported by data and asserts that rigorous quantitative scientific research is sometimes understood as what constitutes evidence. However, Davies [36] expands this definition of evidence by providing seven additional classifications.

Davies' [36] classifications include *statistical evidence*, *descriptive and experiential evidence*, *individual evaluations and research studies*, *implementation evidence*, *ethical evidence*, *economic and econometric*, and *research synthesis*. In this study, the first four categories of evidence were the most relevant. *Statistical evidence* is defined by Davies as evidence that is obtained from official statistics, administrative data, and surveys and describes the nature, dynamics, and size of the issue in question [36]. *Descriptive and experiential evidence* is defined as intuitive or "tacit" knowledge and the lived experiences that are used to shed light on the nature, dynamics, and scale of an issue. *Individual evaluations and research studies* is defined as consisting of desktop research, and *implementation evidence* is defined as documentation of the ways in which obstacles to the adoption of comparable policies have been overcome and if such policies were successfully implemented. These are the four categories of evidence that emerged from the literature selected.

## Results

### Overview of the literature

**Summary of Findings.** The focus of this scoping review was evidence-based decision-making in respect of disability as defined in the UNCRPD which guided our selection and inclusion of papers. After the removal of 19 duplicates, the

search strategy produced a total of 3921 records. Following careful review of titles and abstracts, we screened 141 full texts and excluded papers that did not meet our inclusion criteria. As a result, 16 papers were found eligible for final analysis. We used the Preferred Reporting Items for Systematic Reviews and Meta-Analyses Extension for Scoping Reviews (PRISMA-ScR) guidelines to improve the quality of our methodology and reporting (Fig 1).

Characteristics of records included in this review are summarised in the 16 records shown in Table 1 below, focusing on papers that have a core focus on disability.

For this scoping review, academics were not included as a population of interest. We acknowledge that though academics may be influential to the decision-making process of disability-inclusive initiatives, they do not wield the power to make decisions concerning policy development and programming, neither in government nor NGOs. Table 2 below

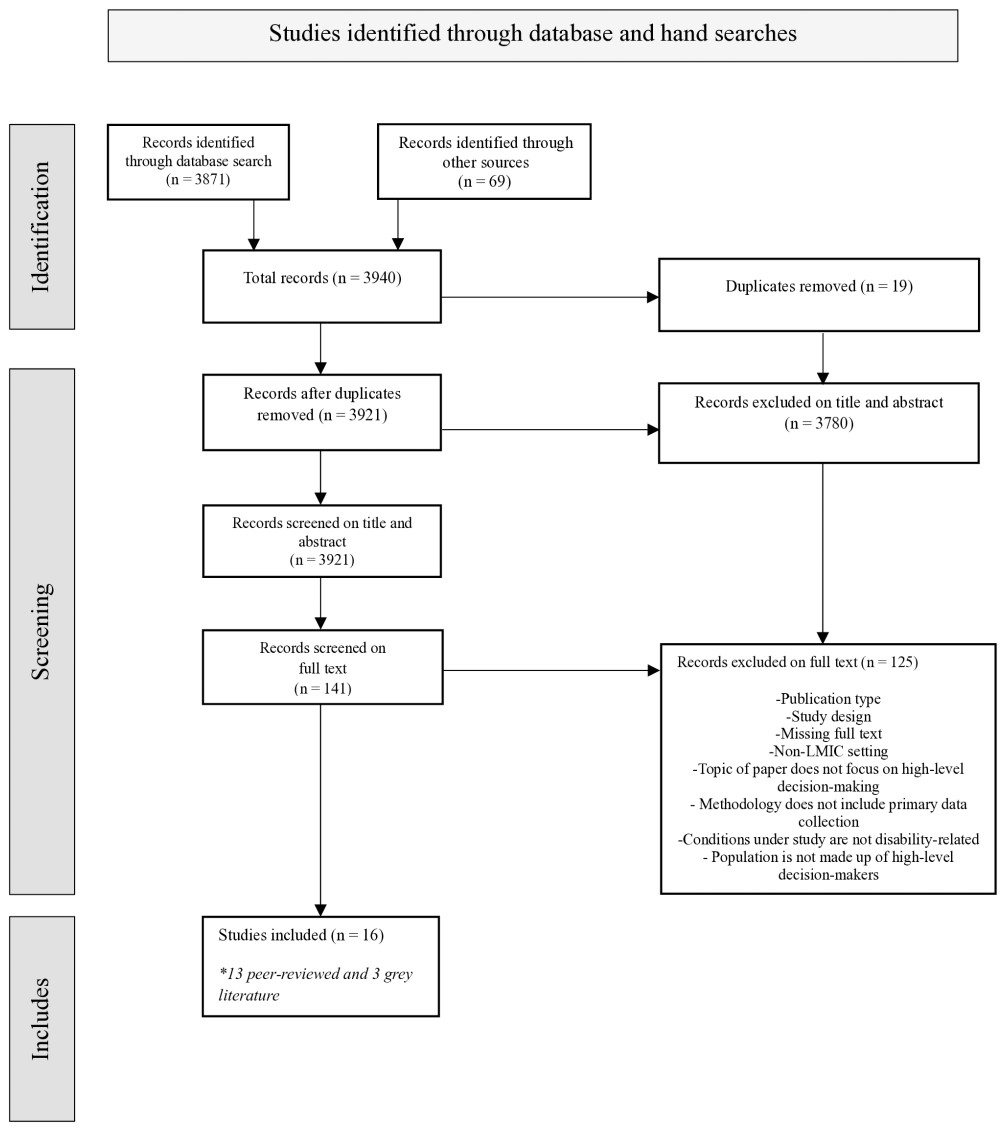

**Fig 1. PRISMA-ScR.**

**Table 1. Summary of included literature.**

| Analysis category | First author & publication date | Type of literature | Stakeholder type (state, non-state, or both) | Sector(s) of participants | Impairment or health condition | Country or countries | Data collection methods | Sample size |
|---|---|---|---|---|---|---|---|---|
| Disability | Braun (2022) | Peer-reviewed | Both | Education | Childhood disability (general) | Tanzania | In-depth interviews (IDIs) | 18 |
| Disability | Brydges (2020) | Peer-reviewed | Both | Education; Health; Social Development; Other (Livelihoods) | Multiple disabilities | Uganda | Questionnaire; IDIs | 25 |
| Disability | Chibaya (2022) | Peer-reviewed | Both | Education; Health; Social Development; Other (Justice; Disability Affairs; Arts and Culture; advocacy organisations) | Multiple disabilities | Namibia | IDIs | 14 |
| Disability | Cleaver (2020) | Peer-reviewed | Both | Health; Social Development | Multiple disabilities | Zambia | IDIs | 24 |
| Disability | Grimes (2021) | Grey | Both | Education | Childhood disability (general) | Afghanistan, Bangladesh, Bhutan, India, Maldives, Nepal, Pakistan, Sri Lanka | Questionnaire; document review; IDIs | Not specified |
| Disability | Heidari (2021) | Peer-reviewed | Both | Health | Neurodevelopmental disability (Phenylketonuria) | Iran | IDIs; document review | 38 (incl. unspecified number of academics) |
| Disability | International Disability Alliance (2017) | Grey | Non-state | Social Development; UN Agencies | Multiple disabilities | Ethiopia, Kenya, Nigeria, Togo, Bangladesh, India, Indonesia, Denmark, Italy, Sweden, Argentina, El Salvador, Peru | Questionnaire | Not specified |
| Disability | Jerwanska (2023) | Peer-reviewed | Both | Health; Social Development; Rehabilitation | Physical disabilities (rehabilitation) | Sierra Leone | Focus group discussions (FGDs) | 37 |
| Disability | Liechtenstein (2022) | Grey | Both | Other (humanitarian) | Multiple disabilities | Somalia | IDIs; document reviews; stakeholder analysis | 9 |
| Disability | Lyra (2023) | Peer-reviewed | Both | Health | Multiple disabilities | Brazil | Document review; IDIs | 7 |
| Disability | Morone (2014) | Peer-reviewed | State | Health | Sensory disability (visual impairment) | 82 LMICs in Africa, America, Eastern Mediterranean, South East Asia, and the West Pacific | Questionnaire | Not specified |
| Disability | Najafi (2021) | Peer-reviewed | Both | Health, Education | Multiple disabilities | Iran | IDIs | 21 |
| Disability | Neill (2023) | Peer-reviewed | Both | Health | Multiple disabilities (rehabilitation) | 47 LMICs and HICs across all WHO regions | IDIs; document review | 65 |
| Disability | Pillay (2022) | Peer-reviewed | State | Education; Health; Social Development | Neurodivergence (autism) | South Africa | IDIs | 6 |
| Disability | Shahabi (2020) | Peer-reviewed | State | Health | Multiple disabilities (rehabilitation) | Iran | IDIs; questionnaire | 36 |
| Disability | Wilbur (2021) | Peer-reviewed | Both | Education, Health, Social Development, Other (Federal Affairs; sexual and reproductive health) | Multiple disabilities (impaired self-care and activities of daily living) | Nepal | IDIs | 12 |

**Table 2. Overview of literature characteristics.**

| Records (n =) | | |
|---|---|---|
| Total number of records | | 16 |
| Overall publication timeline (year - year) | | 2014 - 2023 |
| Top research country research sites (n =) | | |
| Iran | | 3 |
| Nepal | | 2 |
| India | | 2 |
| Literature type | Peer reviewed journals | Grey literature |
| (n =) | 13 | 3 |
| Research designs (n =) | | |
| Qualitative | | 12 |
| Quantitative | | 1 |
| Mixed methods | | 3 |
| Data collection methods (n =) | | |
| Most common | In-depth interviews | 13 |
| | Questionnaires | 5 |
| | Document reviews | 5 |
| Least common | Focus Group Discussions | 1 |
| | Stakeholder analysis | 1 |
| Participants (n=) | | |
| Members of state | | 3 |
| Non-state members | | 1 |
| State and non-state combined | | 12 |
| Sector(s) (n=) | | |
| Most common | | |
| Health | | 12 |
| Levels of Operation (n=) | | |
| Most common | National | 14 |
| Least common | Local | 2 |

presents a characteristic overview of the literature included, specifically focusing on the most and least occurring characteristics across all 16 records.

## In-depth analysis of the literature

**Summary of findings.** We identified 16 papers that focused on disability-related programming, policy-making, or both, and examined them in depth. From the 16 records, we identified 5 commonly occurring aims in the literature (Please see Table 3 below). Overall, the main aims of most of the included literature considered the topic of policy *implementation* over policy *formulation*.

The following sections describe the various types of evidence reportedly used in decision-making and details the barriers to, and facilitators of, the use of each of these types of evidence. We also provide insight into other influences besides formal evidence on decision-making in disability policy and programming processes cited by study participants in these records.

Of the 16 papers, just over half of the records analysed (n = 9) documented the use of more than one type of evidence source [38–45]. Aside from evidence, a range of other influences appeared to significantly shape decision-making in LMIC contexts (see Fig 2). We examined all papers in this review for key influences on disability-related decision-making and

identified 19 "other influences" in total. From these 19 influences, 5 major themes emerged, namely 1) government legislation, power dynamics and involvement (in decision-making), 2) stakeholder relationships, 3) local landscape, 4) funding, and 5) attitudes towards disability. These themes will be further explored in the subsequent sections to provide a comprehensive understanding of the factors influencing disability-related policy and programming decisions.

### Sources of evidence and barriers and facilitators to use of evidence sources

**Statistical evidence.** Approximately 44 percent of papers (n = 7) showed that key stakeholders referenced this source as important information for disability-related decision-making. Jerwanska et al. [41, p. 1799], for instance, reported that key stakeholders with influence on decision-making would "not accept the prevalence data on disability according to the national census conducted" because they considered the information to be inaccurate. Similarly, Liechtenstein [42] recounted the criticisms of key stakeholders who learned that reliable census or disaggregated data to guide resource distribution and policy-making, in work concerning children with disabilities, was lacking overall. Additionally, although cited only once, a notable barrier was the costliness of data collection tools, specifically in terms of expanding the number of census questions to adequately identify and quantify disability prevalence [46].

**Table 3. Summary of disability-related study aims.**

| Commonly occurring aims (n =) | |
| --- | --- |
| Operationalisation of global approaches or commitments in LMIC settings (e.g., community-based rehabilitation (CBR) or the UNCRPD) | 9 |
| Exploring policy implementation barriers | 7 |
| Identifying factors contributing to conducive environments for disability policy-making and programming | 7 |
| Exploring the process of policy development | 6 |
| Documentation of the role of collaboration among multiple key stakeholders | 4 |

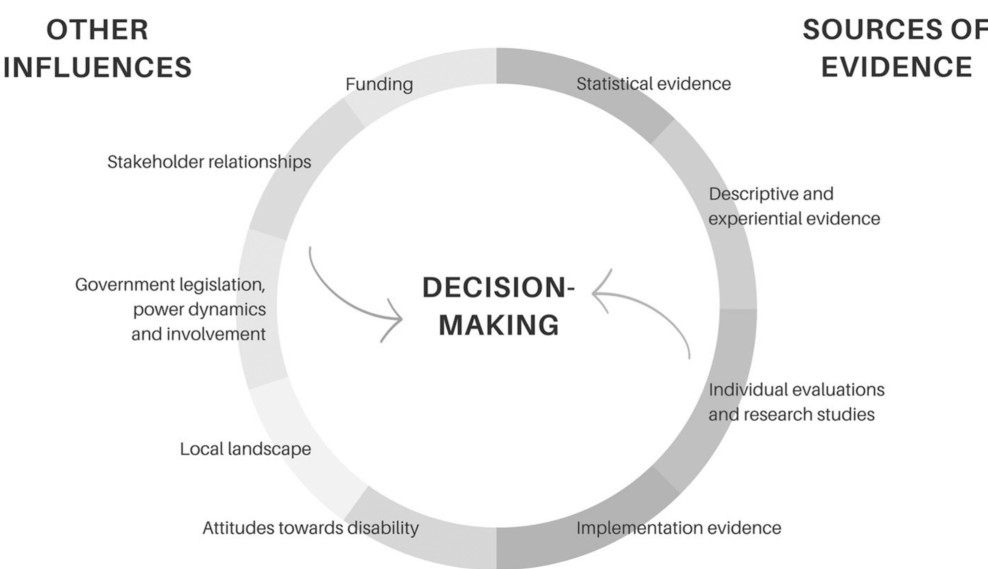

**Fig 2. Disability-focused decision-making in LMIC contexts.**

Regarding facilitators of the use of statical evidence in decision-making identified in the literature, Wilbur et al., for example, reported on the barriers to menstrual hygiene management (MHM) faced by people with disabilities and reflected on the fact that key stakeholders felt that "investment is required to generate evidence" [45, p. 2]. This highlights the role that resources, such as finances, play in the collection of statistical evidence. The capacitation of data capturers was also cited as important for ensuring that the quality of data collected by censuses was adequate. [41].

Additionally, stakeholders noted that the strengthening of electronic information management systems could facilitate the increased trust in, and use of, statistical evidence in some countries, because stronger information systems would support "ongoing improvements in identifying disability, data collection, monitoring and quality assurance" [38, p. 6]. Finally, stakeholders reported that the accuracy and reliability of evidence could be improved by developing and disseminating standardised methods for collecting and analysing disability-related data. This was detailed by Grimes et al., who further asserted that if the Washington Group on Disability Statistics (WG) questions were used across all data collection efforts, the "generation of consistent and comparable data" would be facilitated, supporting its use as evidence in policy and programming [38, p. 6].

**Descriptive and experiential evidence.** Within the disability-focused literature analysed, we characterised descriptive and experiential evidence by the theme *disability representation*. We defined this theme as the experiences of people with disabilities, and those of their families and caregivers, who are self-representative and do not formally constitute associations or organisations but are able to assist in gathering research evidence and hold tacit knowledge that may be useful to key stakeholders in high-level decision-making. This theme included two categories: 1) *lived experiences,* and 2) *formally recognised disability advisors*.

An example of *lived experiences,* as a form of evidence, can be seen as documented by Grimes et al. where the authors described "the practice of consulting and involving children with disabilities, their families and caregivers in decision-making processes" [38, p. 7]. The authors found that the incorporation of this type of evidence was limited at local and national levels of policy-making and programming but was nonetheless perceived as relevant and necessary in efforts toward the development of inclusive education.

We created the coding category, *formally recognised disability advisors,* to refer to disability representative bodies, such as organisations of persons with disabilities (OPDs) (also known as disabled people's organisations or DPOs), disability focal point persons (FPPs), expert disability advisors, and the medical associations and national committees that collaborate with policy-makers on disability-related matters. This source of evidence was cited in almost half of all records. The representative bodies described by participants in this category were either formalised structures within governing bodies or independent, non-governmental bodies (NGOs). Formally-recognised disability advisors fulfilled tasks such as consultations with government departments, policy formulation, tool development, primary or secondary scientific data collection, advocacy, or report-writing [42–45,47,48]. Jerwanska et al. [41] specifically noted the exclusion of people with disabilities from dialogues around policy-making for disability. This exclusion appeared to counteract the efforts of disability representative bodies in being a useful source of evidence for decision-makers.

Insufficient scientific knowledge and skills among members of formally recognised disability advisory bodies, and weak collaborative capacity between government sectors and these advisory bodies commonly led to their inability to influence decision-making. Moreover, the poor structure and capacity of disability advisory bodies in terms of policy-related technical skills appeared to result in less influence in decision-making spaces. As an example, Cleaver et al. reported that FPPs did not have sufficient power because they were "ministry employees in low level positions who were unable to access ministry information or influence decisions" [47, p. 5], owing to a lack of detail provided in a national strategic plan regarding their function. Additionally, insufficient funding and the lack of shared identity and coalition among people with disabilities was seen to reduce their influence on policy, as it inhibited their ability to engage politically influential actors [44].

**Individual evaluations and research studies.** An additional source of evidence observed in this review was desk research. Under this code, we included papers (n = 2) that cited the use of secondary data. For example, Heidari et al. [39]

reported that research from other countries was used to inform the adaptation and development of international standards for a novel phenylketonuria (PKU) screening programme in Iran.

Barriers that participants cited against the use of evaluation and research study evidence included scientific data that was inconsistent and unreliable, and the invisibility/limited accessibility of existing resources, particularly published data [40]. However, the availability of scientific human resources (such as academic and medical personnel), the increased awareness and capacity of key stakeholders in disability-related research matters, and the establishment of collaborative relationships between researchers and key decision-making stakeholders were all cited as factors that facilitated this use of evidence [39,40,43,49]. Moreover, pressure from relevant stakeholders (like OPDs, people with disabilities working with government and civil society organisations (CSOs), and NGOs external to government) and the consistent reporting of methodologically-sound, disability-disaggregated data [39,40] were also evenly split as cited facilitators of the use of this kind of evidence.

**Implementation evidence.** The next most common source of evidence reported in reviewed records was classified as *implementation evidence* which includes international guidelines (n = 2). Examples of sources reportedly used by key stakeholders were World Health Organization (WHO) guidelines, specifically those focused on rehabilitation and Universal Health Coverage (UHC), and WHO checklists for (health and other) system assessments [44,50].

Barriers to the use of implementation evidence include influential key stakeholders' refusal to adopt international recommendations if they considered these irrelevant to their local context. Brydges et al. [51] also noted that participants had commented on the insufficient human resources and funding available in LMIC settings, meaning that adherence to international guidelines can result in a dependence on volunteers who may later leave. Additionally, the insufficient transfer of technical knowledge from international to national and sub-national actors was observed as another hindrance to implementation in LMICs. Within a sample of influential key stakeholders who stated that they knew about the CBR approach, for example, "70% said that they were unfamiliar with the CBR *guidelines,* which provide concrete guidance on how to implement a CBR [programme]" [51, p. 1607]. The limited sustainability of programming influenced by international guidelines or principles due to changing political circumstances, such as the withdrawal of international support because of shifts in donor priorities, for example, also affects the likelihood that LMIC-based decision-makers will adopt international guidelines as a source of evidence [44,51]. These findings suggest a notable trend toward a preference for using local population-level data as evidence in the process of high-level decision-making.

Conversely, strong collaborative relationships between multisectoral national and sub-national actors, and the easy availability of international guidance—such as WHO guidelines as a rationale for promoting rehabilitation's integration into the health system, for instance [44]—could facilitate the incorporation of implementation evidence into decision-making. Additional facilitators in this area included the active involvement of important stakeholders, such as government leaders who champion disability-related matters in political arenas, good governance, and the availability of standardised tools. A summary of the aforementioned barriers and facilitators to the use of specific forms of evidence is illustrated below (Fig 3 and Fig 4).

## Other influences on decision-making

Next, we detail each identified influence on decision-making, outside of evidence, reported in the literature analysed (Fig 5) in turn, providing examples from the literature.

**Government legislation, power dynamics and involvement.** The first influence on decision-making identified outside of evidence was themed "government legislation, power dynamics and involvement". This theme consisted of categories such as legal frameworks, government involvement, policy windows, power dynamics due to hierarchical political structures, and national prioritisation (or de-prioritisation) of disability-related matters.

In terms of the influence of *legal frameworks*, Pillay et al. [52] and Wilbur et al. [45] noted that a national constitution can be used to mandate policy development in support of people with disabilities and uphold their rights as citizens. Wilbur et al. [45] argued that this was rooted in a state's desire to maintain the identity of a country committed to equality.

## Population-level data

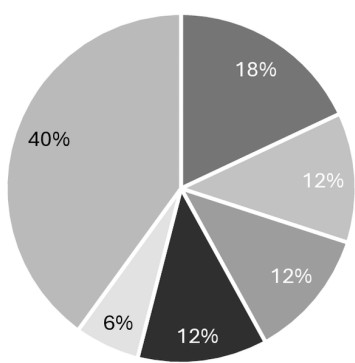

- Inaccurate/underestimated scientific data
- Inconsistent or outdated definitions of disability
- Research applicability
- Insufficient scientific knowledge, experience and expertise
- Costliness of data collection tools
- Miscellaneous

## Disability representation

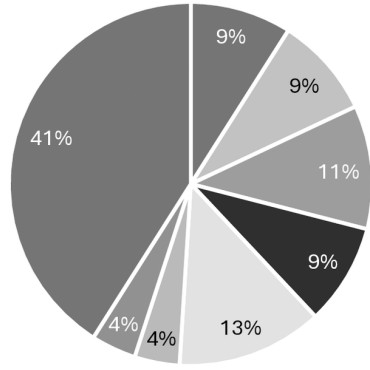

- Insufficient scientific knowledge and skills
- Weak collaborative capacity
- Low prioritisation or profile of disability
- Poor structure and capacity of disability advisory bodies
- Lack of consultation/inclusion of relevant stakeholders
- Insufficient funding
- Lack of coalition among people with disabilities
- Miscellaneous

## International guidelines

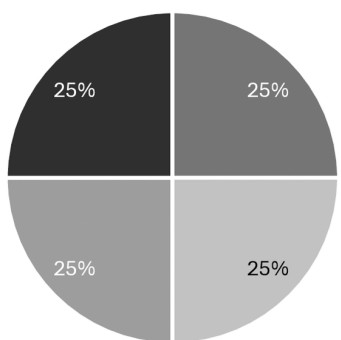

- Local applicability
- Insufficient technical knowledge
- Insufficient human resources
- Changing political context

## Desktop research

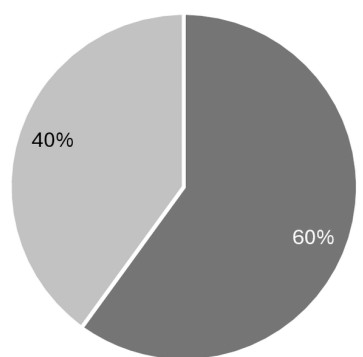

- Inconsistent and unreliable scientific data
- Invisibility/limited accessibility of published data

**Fig 3. Most frequently cited barriers to sources of evidence used in decision-making.**

## Population-level data

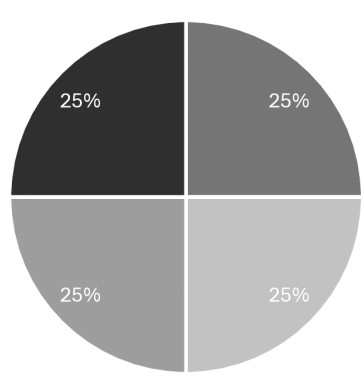

- ■ Financial resources
- ■ Capacitating data capturers
- ■ Strong information management systems
- ■ Standardised data collection methods

## Disability representation

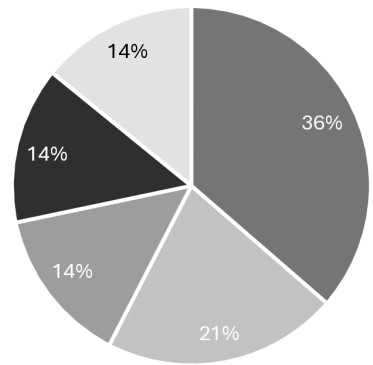

- ■ Collaborative relationships
- ■ Stakeholders' increased awareness and capacity
- ■ Availability of financial resources
- ■ Improved disability-inclusion
- ■ Political will

## International guidelines

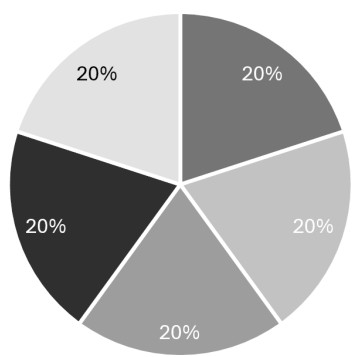

- ■ Collaborative relationships
- ■ International influence
- ■ Involvement of relevant stakeholders
- ■ Good government leadership
- ■ Availability of standardised tools

## Desktop Research

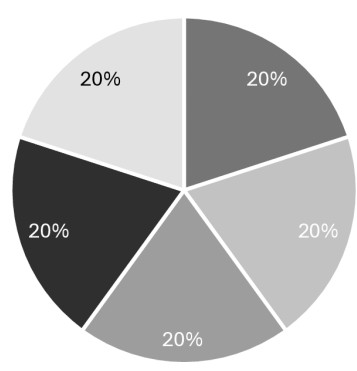

- ■ Availability of scientific human resources
- ■ Awareness and capacity of key stakeholders
- ■ Collaborative relationships
- ■ High quality disaggregated data
- ■ Stakeholder pressure

**Fig 4. Most frequently cited facilitators to sources of evidence used in decision-making.**

*Policy windows*—where an unexpected opportunity occurs in the process of policy development that allows advocates to draw attention to their issues—had a notable influence on disability-related policy and programming within included studies, because these windows generated opportunities for shaping decision-making. Policy windows were reportedly

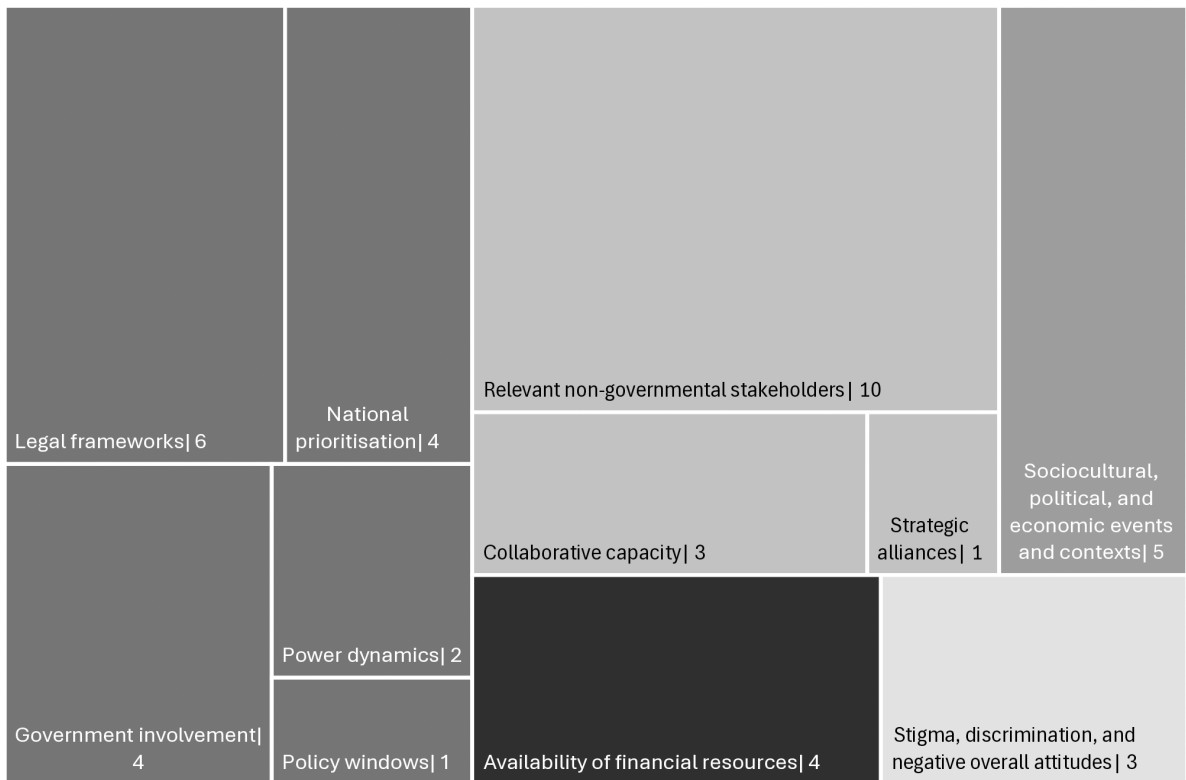

**Fig 5. Hierarchy chart representing other influences on decision-making.**

especially significant for awareness-raising and advocacy [44]. For example, Heidari et al. [39] described the important role of an individual who was able to convince policy-makers to create a policy window, which later led to the development and launch of a successful PKU screening programme in Iran. Conversely, the inability to motivate for prioritisation in agenda-setting negatively affected the progression of disability-related initiatives [44].

Closely related to the idea of "policy windows" are the complex *power dynamics* inherent in many governments due to deeply hierarchical political structures and *national prioritisation*. Though these categories are distinct, there was a notable overlap of these themes in the records reviewed. One study specifically cited the amplified roles of policy-making actors, particularly at the national level [48]. Similarly, another study [49] reported that national governing bodies carried immense influence, and unless government was interested in, and took responsibility for, disability inclusion, disability-related programmes did not receive sufficient support. This lack of support, especially in financial terms, negatively affected the sustainability of programmes for disability inclusion.

With regard to the influence that *government involvement* had on decision-making, this was reportedly either very helpful for, or a hindrance to, the development of disability policies or initiatives. Liechtenstein stated that "to really have a successful disability inclusion project, it has to be owned by the system, so by the government, by the local authorities" because the government "has the power to either jeopardise a project's success", or to help it along [42, p. 11]. In a different study, the national Ministry of Health and Sanitation (with the support of an international organisation) held the most significant power in shaping strategies and policies [43]. Similarly, shifting the responsibility for health and rehabilitation services from a different ministry—the Ministry of Social Welfare, Gender and Children's Affairs (MSWGCA)—to the national Ministry of Health and Sanitation yielded improvements for people with disabilities [41].

**Stakeholder relationships.** Stakeholder relationships were another influence on decision-making. This theme consisted of categories including the *involvement of relevant non-governmental stakeholders*, *strategic alliances*, and *collaborative capacity*. *Strategic alliances* reportedly happened between non-governmental stakeholders, organisations, and agencies. In a study concerning the promotion of eye health in LMIC settings, it was reported that since "local stakeholders are key for securing policy support, specific actions are needed to promote their participation" [43, p. 195], alongside the encouragement of local community stakeholder and local NGO engagement through capacity-building initiatives. These alliances took the form of partnerships between local business communities and international development agencies, as well as collaborations between OPDs, policy-makers, and implementers [53].

Morone et al. [43] noted the significant role that global stakeholders, such as foreign investors and international donors, played in international cooperation, particularly in terms of exerting "positive" (i.e., toward more disability inclusion) pressure on policy-makers. According to Morone et al. [43], the *involvement of relevant stakeholders* such as the WHO, volunteer organisations, and foundations were impactful on decision-making for national policies for blindness prevention in low-income countries (LICs). However, this did not necessarily translate for LMICs and upper-middle income countries (UMIC), where the private sector played a more influential role in this regard [43].

The presence or absence of *collaborative capacity* also influenced decision-making. Neill et al. [44] noted that the capacity to influence decision-makers is strongest when there is a shared goal [or] agenda, highlighting the importance of collaborative abilities and alignment between different groups. In the process of mapping potential stakeholders in physical rehabilitation financing in Iran, Shahabi et al. [49] noted that although some entities could potentially be influential in decision-making spaces (such as the WHO, national welfare organisations, and NGOs), these groups had few opportunities to work together and were therefore less influential in decision-making.

**Local landscape.** The theme "local landscape" considers the sociocultural, political, and economic events and contexts that had an influence on decision-making, according to participants in included studies. Examples of these were "post-conflict political influences" (e.g., dependence on international aid organisations) [51], massive disease outbreaks that happened in previous years [39], international guidelines and global trends that improved the process of implementation for disability inclusion [42], and the political context in general. An example of a political contextual factor was the ratification of the UNCRPD by Brazil in 2008, which Lyra et al. [48] noted had provided an important stimulus for the uptake of disability-inclusive policies. In addition, a strong disability-related social movement played a significant role in bringing about the 2002 adoption of the national health policy by advocating for inclusion and the right to health. However, once political circumstances changed and a fiscally and socially conservative governing party came into power, policies and programming relevant to disability lagged, further demonstrating the importance of the local political context on progress toward inclusion.

**Funding.** The theme "funding" considers the availability of financial resources in support of disability policy or programming. In the context of inclusive education reforms, for example, Braun [50] noted the considerable expense that the establishment and continuation of teacher training programmes carried, which required substantial financial investment in infrastructure and human capital but more importantly, political commitment. Jerwanska et al. [41] noted the negative impact that insufficient government funding had had on health service delivery. Brydges et al. [51] and Neill et al.

[44] similarly referenced the unsustainable dependence on international funding for rehabilitation services that were based on WHO guidelines but were being implemented in low-resource settings.

**Attitudes towards disability.** This theme concerns stigma, discrimination, and negative overall attitudes and beliefs towards disability, and how these have influenced decision-making. Grimes et al. [38] noted that the discrimination and stigma held against children with disabilities could influence not only learning environments and services but just as importantly, policy and programming. Lastly, Braun [50] noted that policy actors reported that individuals within, and beyond, the education system often carry unhelpful and uninformed attitudes, such as the belief that students with disabilities should be kept in the home instead of being allowed to receive an education, which decrease the chances of disability-inclusive policy-making and programming.

## Discussion

This scoping review provides a novel evaluation of literature concerning the use of evidence in governmental and non-governmental decision-making processes around disability-inclusion in LMICs. Our findings highlight a significant gap: the insufficient availability of research focused specifically on the *formulation and development* of disability-related policies and programmes. This was observed in only 16 records (3 of which were grey literature) retrieved from an extensive search. Notwithstanding its relatively small size, this body of work yields important insights, notably around the definition and use of evidence in decision-making, the factors outside of evidence that impact decisions, and the key stakeholders involved in, and processes surrounding, the development of disability-related initiatives. The main insights pertain to those sectors most represented in the literature, the documentation of policy and programme development processes, and the main sources of evidence used in, and other influences on, decision-making toward disability inclusion.

Firstly, this review demonstrates that the health sector is most widely represented in the literature on decision-making in relation to disability in LMICs. This is perhaps unsurprising, given the focus of much disability policy and programming in LMICs on rehabilitation [44,54,55]. However, this finding also suggests that disability as a topic is still medicalised, resulting in poorer visibility in other sectors like social protection and education [56]. By extension, as a population, people with disabilities appear to still be viewed from a medicalised perspective. Many public health interventions depend on the intersectoral participation of sectors including labour, welfare and social assistance, housing, transportation, and education for their success [57,58]. We look at policies across a range of different sectors, recognising that addressing disability requires engagement beyond the health sector. Global health actors can learn from the expertise and approaches used by stakeholders in non-health sectors, particularly in areas such as inclusive education, social development, and labour policies, where disability is addressed from a broader rights-based and social inclusion perspective. These sectors often have well-established frameworks for participation, equity, and accessibility that could inform global health strategies and enhance their effectiveness. Considering the fact that disability affects individuals in diverse and varied ways, but in all aspects of their lives, it is important that intersectoral collaboration is an early and central aspect of any disability-related programme or policy. This is particularly relevant to global health, which relies on multisectoral cooperation to address the broader determinants of health. Strengthening intersectoral collaboration through disability policy and programming can serve as a model for global health interventions, demonstrating how partnerships across sectors can enhance equity and sustainability.

The evaluation of disability-inclusive policies and programmes could be a promising site for promoting intersectoral collaboration because evaluative processes provide numerous opportunities to bring together stakeholders with specific roles in the implementation of a policy or programme, and whose interests are impacted by it, both during the planning stages and during the evaluative process itself. Evaluations can be planned as deliberate procedures in which stakeholders from many sectors consider and balance differing viewpoints, agree on the main subject of the assessment, and decide which outcomes to measure [57]. This approach would be beneficial for the development of disability policy and programming in LMICs and offers valuable lessons for global health more broadly. Integrating insights from non-health sectors into global

health frameworks could lead to more comprehensive and sustainable approaches to disability inclusion and health equity on a global scale.

Secondly, the literature identified in this review explores policy and programme implementation, or the lack thereof, as opposed to policy *formulation*. An optimistic interpretation of this finding is that it suggests that many LMICs have existing disability policies to be reflected upon. However, the failure to document policy and programme development processes means that the presence or absence of different types of evidence—and the process of how decisions are made—remains opaque. To foster better knowledge and more informed decision-making among those in charge of government operations, evaluation experts are tasked with providing retrospective evaluations of the execution, output, and result of government actions [59]. Evaluation entails retrospective analysis in order to steer ahead more effectively. This process assists with separating what is valuable from what is not. Additionally, because evidence develops over time, we need to understand and identify potential evidence "entry points" during policy and programme reviews and updates, in order to improve practices in alignment with emerging knowledge. According to Vedung [59], all forms of assessment are predicated on the notion that views, beliefs, intents and judgements play a significant role in political and administrative activity. It is worth considering and understanding how these factors work, but this can only be achieved through thorough documentation of the policy and programme development processes.

Thirdly, government legislation, dynamics, and interest in disability inclusion were reported as the most powerful influences on the development of progressive disability-related policy and programming. This is likely due to the perceived interplay between the roles of key stakeholders with decision-making power, their knowledge of, and attitudes towards, disability initiatives, and the actions (or a lack thereof) that they take. Gilson and colleagues [60] acknowledge that governance is a multi-layered, intricate process that affects policy enactment, decision-making, and service delivery in the field of health systems research. Thus, "whole-systems" awareness requires knowledge and comprehension of both the "software" (e.g., values, norms, and relationships) and "hardware" (e.g., finances, resources, and infrastructure) aspects of a system [60]. In our review, the national legislation of a country and actions resulting from the involvement of government could make or break programmes in terms of their progress and sustainability. Therefore, disability initiatives can grow or thrive when the state takes responsibility for them and demonstrates good leadership.

Local statistical evidence from censuses and survey data gathered also reportedly played a role in the development of disability policies and initiatives. A preference for local data combined with the lack of uniformity or standardisation of data collection tools nationwide may explain the regular use of WHO guidelines and tools by some influential key stakeholders, and the appraisal of such to be ill-fitting for their local context, by others. Cairney [56] asserts that the framework for policy formation is established beliefs, therefore, any new evidence regarding the efficacy of a policy must be accompanied by a change in emphasis and persuasive arguments. At times, social or economic crises may serve as catalysts to transfer focus from one matter to another, aided by specific forms of evidence. However, substantial policy transformations are said to be uncommon [56].

Where collaboration and strong relationships among relevant stakeholders were reported, particularly between local and international non-governmental bodies, the cumulative influence of pressure from these entities was sometimes enough to sway government decisions. According to literature on policy-making, but with applicability to programming too [56], these kinds of decisions often occur in unpredictable and complex environments. These environments are characterised by the following features: a wide variety of organisations and individuals influencing policy at different levels of government; a multitude of rules and norms observed across different contexts; close connections (or "networks") between policy-makers and influential actors; the prevalence of particular beliefs or "paradigms" in discussions; and situations or events that have the potential to swiftly capture the attention of policy-makers. It seems possible that collaborative efforts among local and international NGOs who have access to, and make use of, higher quality disability data may have the power to significantly influence government decisions in complex multi-level policy environments.

Based on our scoping review findings, we recommend the encouragement of multidisciplinary approaches that integrate the perspectives of varied sectors including (but not limited to) health, education, social sciences, construction, and labour in the work of disability policy and programming. We further advise that governments consistently involve a diverse range of stakeholders, including people with disabilities and advocacy groups, in the early stages of policy and programme formulation to ensure effective inclusivity. Moreover, standardising definitions of disability and data collection tools across LMIC research while also adapting international guidelines to local contexts may help to address inconsistencies in scientific research and data collection tools. Additionally, clear, country-specific implementation strategies, robust monitoring and evaluation frameworks, and indicators disaggregated by age, gender, and disability status are essential for creating useful statistical evidence and encouraging the use of this data in efforts towards the development of disability policy or programming in LMICs. Finally, strengthening leadership, with a focus on both the "software" and "hardware" aspects of governance, is crucial for sustainable progress in disability initiatives.

## Limitations

In the process of conducting this scoping review, we relied on the data obtained and extracted from the literature that met our inclusion criteria. However, we acknowledge that by basing this review solely on papers published in English, we may have lost valuable insights, especially those emerging from Latin or Francophone countries. Finally, due to time constraints and the desire to present succinct findings, we documented only the most frequently occurring and salient barriers to, and facilitators of, evidence use, and other influences on decision-making, and limited these to the top five most commonly occurring themes.

## Conclusion

This scoping review highlights the influences shaping decision-making around disability-related initiatives in LMICs, the types of evidence that inform these initiatives, and the factors that may support or hinder evidence use. Most studies focused on policy and programme implementation rather than formulation, with the health sector featuring most prominently. Government legislation, political dynamics, and interest in disability inclusion emerged as the strongest drivers of progressive disability policies, while local statistical data—particularly from censuses and surveys—also significantly contributed to disability-inclusive decision-making. Collaboration among stakeholders, especially local and international NGOs, sometimes proved sufficient to sway government decisions.

Broadly, this review identifies key barriers and enablers, offering targets for intervention and advocacy. Moreover, the review highlights factors that could enhance disability inclusion in decision-making, particularly through targeted engagement with decision-makers. The powerful influence of government legislation, power dynamics, and (level of) interest in disability inclusion, combined with a reported skepticism of local evidence, suggests that stakeholders may make specific decisions due to insufficient access to, or trust in, the available evidence. This possibility should be explored further in future research. Stakeholders seeking to support practices around evidence-based decision-making in LMICs must be cognisant of these differing framings of the concept of evidence, and the importance of influences outside of evidence on decision-making behaviours, in the development of relevant interventions.

## Supporting information

**S1 File. Preferred reporting items for systematic reviews and meta-analyses extension for Scoping Reviews (PRISMA-ScR) checklist.**
(DOCX)

**S2 File. Search strategy.**
(PDF)

**S3 File. Grey literature and Google Scholar hand search.**
(DOCX)

## Acknowledgments

We would like to thank Vittoria Lutje of the Liverpool School of Hygiene and Tropical Medicine for her support in conducting systematic database searches.

## Author contributions

**Conceptualization:** Xanthe Hunt.

**Data curation:** Ebruphiyo Ruth Useh, Xanthe Hunt.

**Formal analysis:** Ebruphiyo Ruth Useh, Zara Trafford, Prince Changole.

**Funding acquisition:** Xanthe Hunt.

**Investigation:** Ebruphiyo Ruth Useh, Zara Trafford, Prince Changole, Xanthe Hunt.

**Methodology:** Ebruphiyo Ruth Useh, Zara Trafford, Prince Changole, Xanthe Hunt.

**Project administration:** Zara Trafford.

**Resources:** Xanthe Hunt.

**Supervision:** Zara Trafford, Xanthe Hunt.

**Validation:** Zara Trafford, Prince Changole, Xanthe Hunt.

**Visualization:** Ebruphiyo Ruth Useh.

**Writing – original draft:** Ebruphiyo Ruth Useh.

**Writing – review & editing:** Ebruphiyo Ruth Useh, Zara Trafford, Prince Changole, Xanthe Hunt.

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
