## [Decision Letter · Decision Letter 0]

30 Dec 2024

PGPH-D-24-02548

Sources of evidence, pathways, and processes to support disability-inclusive decision- making in low- and middle-income countries: A scoping review

Dear Dr. Useh,

Thank you for submitting your manuscript to PLOS Global Public Health. After careful consideration, we feel that it has merit but does not fully meet PLOS Global Public Health’s publication criteria as it currently stands. Therefore, we invite you to submit a revised version of the manuscript that addresses the points raised during the review process.

The reviewers and I have provided constructive feedback to improve the paper, which mainly involve clarifying method decisions, writing, and anchoring the work on global health scholarship- particularly around intersectoral collaboration and working with stakeholders outside the health sector.  Please carefully consider and respond to the suggestions for improvement. 

We look forward to receiving your revised manuscript.

Kind regards,

Lavanya Vijayasingham, PhD MPH

Academic Editor

Journal Requirements:

1. Thank you for uploading your study's underlying data set. Unfortunately, the repository you have noted in your Data Availability statement does not qualify as an acceptable data repository according to PLOS's standards.

2. Please provide an Author Summary. This should appear in your manuscript between the Abstract (if applicable) and the Introduction, and should be 150–200 words long. The aim should be to make your findings accessible to a wide audience that includes both scientists and non-scientists. Sample summaries can be found on our website under Submission Guidelines:

https://journals.plos.org/globalpublichealth/s/submission-guidelines#loc-parts-of-a-submission

Additional Editor Comments (if provided):

Dear authors,

Thank you for submitting this important work for publication consideration. This work has been reviewed by three reviewers who have provided constructive feedback to strengthen the manuscript. I am recommending a major revision, for the reason that some conceptual details and related methods need to be clarified, along side an overall review of the writing- which feels rather long-winded at the moment. Readers are time-poor, and often need to be pointed towards the important points quickly. Some of your important points are buried in text, and not signposted as the important findings. There are several areas particularly in results section 1 that can be collapsed into tables, and words can be better used to discuss rather than describe.

Please consider the feedback from reviewers carefully and respond to this in your resubmission. Importantly, please review the reference list and check missing details, refer to PRISMA diagram, search strategy and inclusion details more clearly in methods, and re-run the search in line with R1 suggestion to update results. This also aligns with the point around mental health inclusion below.

Additionally, I also have reviewed the paper and annotated some questions and points to clarify as comments in the attached PDF. My main concerns for this work are the framing around global health literature and scholarship, and then the inclusion/ exclusion of mental health in the overall analysis.

1) This is a global health journal, so readers will expect to see some framing around related topics and conversations. It is not clear at the outset if you refer to disability policy beyond the health system- this is an important framing bit that will strengthen the paper. You have included policy outside health which is very pertinent to intersectoral involvement and engagement by health system actors. Once you shorten and tighten the overall writing, please consider including a framing around the importance of intersectoral coordination, collaboration etc- and how that can benefit the goals of global health actors, particularly around disability and health policy. Additionally, in the discussions, please consider reflecting on what health sector actors can learn from the non-health sector stakeholders- as described in the included papers, and how this can also serve intersectoral collaboration work in global health.

2) The inclusion and exclusion of mental health policies is rather confusing. It seems like a matter of convenience rather than a conceptual decision. My suggestion is to either include mental health across all sections of analyses or remove it entirely. The paper will still serve its intended goals with or without mental health inclusion. You can anchor strongly on a conceptual definition of disability, and how disability is often framed in policy within LMICs- you could describe this in the introduction to guide readers. If you decide to remove mental health, you could say something like- in this paper we focus on the case of physical disabilities (for example- please clarify what you include, why this supports the research aim).

Hope this helps, and I look forward to reading the revised and resubmitted version of this important work.

Reviewers' comments:

Reviewer's Responses to Questions

**Comments to the Author**

1. Does this manuscript meet PLOS Global Public Health’s publication criteria ? Is the manuscript technically sound, and do the data support the conclusions? The manuscript must describe methodologically and ethically rigorous research with conclusions that are appropriately drawn based on the data presented.

Reviewer #1: Yes

Reviewer #2: Partly

Reviewer #3: Yes

2. Has the statistical analysis been performed appropriately and rigorously?

Reviewer #1: N/A

Reviewer #2: N/A

Reviewer #3: Yes

3. Have the authors made all data underlying the findings in their manuscript fully available (please refer to the Data Availability Statement at the start of the manuscript PDF file)?

Reviewer #1: Yes

Reviewer #2: Yes

Reviewer #3: Yes

4. Is the manuscript presented in an intelligible fashion and written in standard English?

Reviewer #1: Yes

Reviewer #2: Yes

Reviewer #3: Yes

5. Review Comments to the Author

Reviewer #1: Thank you for the opportunity to review this interesting manuscript. You have presented rich findings that will be useful for applied work in the future. There are comments to address below. Of major thought is consideration on the continued inclusion of the mental health studies. As I recommend below, I suggest you consider including findings from these or extracting into a second paper. They are lost in this paper. Good luck with the amendments and look forward to seeing this at next draft. Additionally, I note that your search was conducted nearly a year ago. I suggest you re-run the search for this year and include any published in 2024. Considering this won't be published by the journal until 2025, your search will be out of date quickly. It shouldn't be a lot of work to add in updated search results.

Abstract

• Include the databases searched

Introduction

• Line 36: Check journal guidance but may need year after ‘Goldman and Pabari’.

• Line 37: You may need to explain what each of these steps include. I don’t intuitively know what ‘intervention selection’ is, for example

• Line 47: This section on the what we mean by evidence is quite long-winded. If you are not addressing this in this study, as you say, then consider removing or condensing to very simple terms. From line 52 to 58 is particularly long-winded and could be refined

• Line 60: Need to spell out LMIC in full first

• Line 65: When describing the lack of disability data, be clearer with what you mean by “insufficient and irregular”

Materials and methods

• Line 119: What do you mean a scoping review methodology? Outline from who and where this methodology has derived. And outline the five steps

• You have not included a clear inclusion criteria. Please add using a clear structure, e.g. PICO or SPIDER. Include as a separate section before the search strategy. Note: I now see something akin to this at the start of the results. It should come in the methods and be more clearly structured, as suggested

• Good array of databases used

• The search for grey literature could be clearer. For each of these websites, how did you search for documentation? Did you search every page or use their search function? Why did you pick these organisations? Suggest also including the full names of organisations, not acronyms

• For the Google Scholar search, how did you ensure this was systematic? What search terms did you use? How did you know when you stop looking (i.e. did you look at the first 10 pages, for example)?

• The search strategy looks fine, although you have not included in the search the names of actual countries classified as LMIC. This should be listed as a limitation, as you may have missed some studies that did not mention the search terms used, and simply named the country, e.g. Zambia without mentioning Africa or lmic or developing

• Line 133: “Worked in one pair to screen the literature”. You can likely delete this here as you explain in the study selection process. I had originally asked for more information here, but saw it came later, so suggest removing at this point, so as not to confuse the reader

• Why did you use the cutoff of 1990?

• Line 135: Your search was conducted nearly a year ago. Consider updating this search to capture data from 2024. This is important, given the increase in disability-related research we are seeing

• Line 157: Make clear by ‘below’ you mean ‘in the results’. I was unclear where Part 1 would be

Results

• In the PRISMA chart, I would expect to see reasons for exclusion at full-text in the relevant box

• Line 298: Add instead to the methods section under data analysis/synthesis as this is not results. Similarly, suggest outlining the details of each category in the methods, rather than at the beginning of each section in the results. The results feel long as is

• Line 331: You have already outlined that you are reviewing the literature for barriers (see line 310). Suggest review of results and methods to cut down on repetition

• The results are informative and well-structured.

• Although I understand your rationale, it does seem a bit off in practice when you have not synthesised any information from the included mental health literature. The results are already very long and I am reluctant to add more, but I do think you need to consider providing a short paragraph on evidence related to mental health studies, under each evidence section. Otherwise why include them at all, except to show that mental health is a more researched area? Feels a wasted opportunity, given the effort to source the studies. If not here, consider providing in a separate paper and excluding from this. You make this note in the methods, to say “The concerns of people with sensory, physical, intellectual, self-care, or communication impairments may differ, in some systematic ways, from those documented in the global mental health literature, and so lessons learned from the latter literature may have limited applicability to the concerns of people with disabilities generally.” Perhaps then exclude them from this paper with this rationale and write another specific to mental health.

• This is a long section and long piece of work generally. I encourage you to consider putting the table or some figures in as supplementary information

Discussion

• Generally a thoughtful and interesting discussion

• Line 681: Could you structure this as a clearer recommendations section and consider bulleting the different recommendations

• Limitations: See notes in the methods section

Reviewer #2: Introduction: The introduction is well-presenting the significance of this study. However, adding literature-based reasoning on which specific sectors, e.g., health, economy, and education, have mostly excluded evidence-based disability-inclusive decision-making and implementation and how this study will contribute to addressing that exclusion will strengthen the introduction.

Method: The method needs to be re-structured. It comprises all the required components but in a scattered and unnecessarily lengthy manner, making readers question its validity.

Authors should consider incorporating the PRISMA guideline in a flow chart form, which will include how many records were identified according to search strategy, how many were left after screening, how many were left after removing duplication, how many full articles were available and finally how many were included for analysis.

The search strategy should be very clear. What were the search words, and how were the words combined by using “AND” and “OR”? This should be explicitly mentioned in the search strategy table added in the supplementary file. However, the inclusion and exclusion criteria should be included in the method with justification. For example, if authors have not included any policy analysis in their study, then why?

The coding categories and their definitions are more suitable to present in the method section, not in the result.

Remaining comments are in the attached word document

Reviewer #3: 1. This is a clear and well-written manuscript and of great relevance in relation to the neglected issue of disability inclusion in low- and middle-income country (LMIC) policies and programmes. It addresses an important knowledge gap and helpfully unpacks the piecemeal rather than systematic way in which policies and programmes are shaped, both because of a dearth of LMIC evidence and because where evidence does exist it may not be considered suitable or may not be drawn upon in a systematic way. Although this is not a central focus of the study, the authors also note that what constitutes evidence is itself not fully articulated, and that in the case of LMICs even basic data such as demographics and specific disability information tends to be highly lacking, compared with high-income countries (HICs). Thus, they acknowledge that evidence-based decision-making models developed in HICs may thus be difficult to apply in LMICs.

2. The authors have dealt carefully with the preponderance of mental health literature and provide a justification for how they did so. They provide a clear and reader-friendly description of their study selection, data extraction and data analysis processes, helpful to potential future researchers and creating a strong sense of trustworthiness and credibility.

3. Table 1 gives a clear and thorough overview of the selected studies but I suggest that in lines 228-230 mention is made of the first 16 studies (with a core focus on disability) being shown in green and the last 22 (with a core focus on mental health) being shown in blue. With this having not been mentioned overtly, and due to the studies not being numbered, the blue took me by surprise and caused me to first look back at the text for clarification and then count the green studies to sure that the colours had been applied in the way I surmised.

4. The results section is well set out, clear to follow and provides additional trustworthiness elements such as explanations of how coding categories were developed and defined. Figures 2, 3, 4 and 5 are bold and succinct, but I do note that the use of colour can be problematic both for people with colour blindness (ironic, given the focus of the paper) and for reproduction of the article in black and white. The latter is unfortunately likely to happen in lower-resourced settings, reducing its value to stakeholders like policymakers and civil society organisations.

5. The discussion section flows well, integrating the results and framing them against the literature before making some pertinent recommendations.

6. The conclusion is somewhat long but does not suffer from repetition, and making it more concise is a nice-to-have rather than must-have.

7. There are a few language and typographical issues to address:

a) In lines 265 and 267 I looked twice at the terminology "worked with", which seems strange - it should perhaps be replaced with something like "focused on", to more aptly reflect the authors' roles in the relevant studies/sources.

b) In line 283 there is sudden use of a fractional percentage ("8.50%") whereas for the most part whole numbers are used and I presume the authors thus rounded off decimal points. For reader-friendliness, I suggest applying whole numbers consistently throughout the manuscript.

c) Although I spotted almost no typos, one in line 456 (in the word "insufficient") did jump out at me.

8. Finally, I have one big concern and that is the reference list. Too many references fail to provide enough information for them to be reliably located by readers (eg: some do not provide the name of the institution/organisation which produced them, and many do not provide a URL even though one is almost certainly available and could have been readily found and included by the authors, as a service to their readership). Particular attention needs to be paid to refs 1, 4, 8, 13, 15, 21, 29 and 46 in this regard, and it is especially important to ensure that the grey literature is findable. Ref 8 would benefit from adding capitals (Global Governance Programme & School of Transnational Governance) and should specify where this school is located. Ref 15 does not indicate which organisation produced that study/source, while ref 29 gives neither that sort of detail nor provides the source's date. Fixing up the reference list will make it match the generally impeccable writing style, attention to detail and reader-friendliness of the rest of the manuscript.

6. PLOS authors have the option to publish the peer review history of their article (what does this mean? ). If published, this will include your full peer review and any attached files.

**Do you want your identity to be public for this peer review?** For information about this choice, including consent withdrawal, please see our Privacy Policy .

Reviewer #1: No

Reviewer #2: **Yes: ** Shahpara Nawaz

Reviewer #3: No

---

## [Editor Report · Decision Letter 1]

26 Mar 2025

PGPH-D-24-02548R1

Sources of evidence, pathways, and processes to support disability-inclusive decision- making in low- and middle-income countries: A scoping review

Dear Dr. Useh,

Thank you for submitting your manuscript to PLOS Global Public Health. After careful consideration, we feel that it has merit but does not fully meet PLOS Global Public Health’s publication criteria as it currently stands. Therefore, we invite you to submit a revised version of the manuscript that addresses the points raised during the review process.

We look forward to receiving your revised manuscript.

Kind regards,

Lavanya Vijayasingham, PhD MPH

Academic Editor

Journal Requirements:

Additional Editor Comments (if provided):

Thank you for comprehensively addressing the review comments and feedback. The paper reads much stronger now.

I am recommending a minor but important edit:

1. requesting a more prominent framing and specific mention of global health (instead of international development) in the introduction, with relevant global health references.

2. please clarify and state if you are studying intersectoral disability policy and link this back to how this is crucial for global health and the health outcomes of people with disabilities in the introduction.
---

## [Editor Report · Decision Letter 2]

9 Apr 2025

Sources of evidence, pathways, and processes to support disability-inclusive decision- making in low- and middle-income countries: A scoping review

PGPH-D-24-02548R2

Dear Miss Useh,

We are pleased to inform you that your manuscript 'Sources of evidence, pathways, and processes to support disability-inclusive decision- making in low- and middle-income countries: A scoping review' has been provisionally accepted for publication in PLOS Global Public Health.

Best regards,

Lavanya Vijayasingham, PhD MPH

Academic Editor

Thank you for compressively addressing the comments and suggestions through this review process. I am happy to accept this revised version of the manuscript for publication.

Congratulations, and wishing you the very best research and publication journeys.